nanotechnology/organic chemistry/ materials science

nano/microfibre aerogels, electrospinning, solvent-vapour treatment, air filtration

**Author for correspondence:**
Bingyao Deng
e-mail: bydeng@jiangnan.edu.cn

This article has been edited by the Royal Society of Chemistry, including the commissioning, peer review process and editorial aspects up to the point of acceptance.

# Robust polyimide nano/microfibre aerogels welded by solvent-vapour for environmental applications

Ying Shen[1], Dawei Li[1], Bingyao Deng[1], Qingsheng Liu[1], Huizhong Liu[1] and Tong Wu[2]

[1]Key Laboratory of Eco-Textiles (Ministry of Education), Nonwoven Technology Laboratory, Jiangnan University, Wuxi 214122, China
[2]The Wallace H. Coulter Department of Biomedical Engineering, Georgia Institute of Technology and Emory University, Atlanta, GA 30332, USA

YS, 0000-0002-2472-1844; BD, 0000-0001-5652-3503

Due to the high porosity, resilience and ultra-low density, polymer nanofibre-derived aerogels (NFAs) have been widely investigated in recent years. However, welding of the fibrous networks of NFAs, which has been proved extremely essential to their structural performance, still remains a major challenge. Herein, electrospun polyimide (PI) nano/microfibres were used as building blocks to construct hierarchically porous aerogels through a solid-templating technique. By further welding the adjacent nano/microfibres at their cross-points in a controllable fashion by solvent-vapour, super elasticity was achieved for the aerogels, with a recoverable ultimate strain of 80%. It is noteworthy that this process is free from cross-linking, heating and significant structure changing (i.e. chemical structure, crystallinity and fibrous network). Additionally, the porous structure of PI nano/microfibre aerogels (PI-N/MFAs) could be tuned by adjusting the organization of microfibres from a disordered/ordered cellular to a uniform structure. The as-obtained aerogels showed ultra-low density ($4.81 \text{ mg cm}^{-3}$), high porosity (99.66%), and comparable or higher recoverable compressive strain and stress relative to the other nanofibre-based aerogels. Furthermore, we showed the potential of such an aerogel for particle or aerosol filtration. PI nanofibre aerogels composite filters (PI-NFACFs) manifested excellent performance in $PM_{2.0}$ filtration (99.6% filtration efficiency with 115 Pa pressure drop). Therefore, this study brought a new perspective on the simple preparation of nanofibre-based aerogels for air filtration.

# 1. Introduction

Owing to the hierarchical, three-dimensional (3D) network, ultra-low density and high porosity, aerogels have drawn great attention for applications involving filtration and separation, adsorption, catalyst supports, etc. [1–4]. So far, numerous studies have been carried out to fabricate various aerogels together with free shapes and controllable porous structures. Compared with aerogels made by the traditional sol-gel method, those derived from electrospun nanofibres can avoid the fragility caused by the 'pearl-necklace-like' strings of inorganic nanoparticles and exhibit flexible behaviour [4,5]. Up to now, aerogels have been fabricated from nanofibres made of a variety of polymers such as polyethylene (PE), polyimide (PI) and polyacrylonitrile (PAN), as well as carbon nanofibres [2,6–8]. In a typical process, electrospun nanofibres were cut into short ones (normally in the range of fibre aspect ratio of 80–500) and then dispersed in a proper solvent, forming a uniform suspension. After freeze-drying, the suspension of short nanofibres at different temperatures to remove the volatile solvent, the aggregation was reinforced for the mechanical properties by cross-linking or welding [6,9–12]. Generally, three methods involve the welding of nanofibres to strengthen the fibrous network: heating [6,7,13], chemical cross-linking [8,9] and electromagnetic wave cross-linking [14]. However, major issues still remain in terms of film-like adhesion between fibres, dimensional shrinkage, high energy consumption and operation complexity. It is urgently needed to find a facile and efficient way to synthesize super-elastic, nanofibre-based aerogels without the use of cross-linkers and free of energy consumption. To this end, solvent-vapour treatment will be a good choice to weld nanofibres for mechanical improvement, without causing significant changes to the fibrous structure [15–18].

Platforms made of electrospun nanofibres have been extensively investigated for applications related to air purification due to the interconnected, nanoscale pores, highly specific surface area and fine diameter of nanofibres [19,20]. However, insufficient mechanical strength and high drop pressure caused by the layered stacking of nanofibres limit the purification performance. In this decade, super-elastic and ultra-light nanofibre aerogels have emerged as 3D networks for efficient air filtration [2,21]. Due to the secondary porous structure generated by solvent crystals in the process of freeze-drying, aerogels are ideal as light-weight functional materials for application involving air filtration to capture $PM_{2.5}$. For example, Liu *et al.* [13] fabricated a composite filter consisting of a layer of polypropylene non-woven fabric and another layer of aerogel made by poly(vinyl alcohol-co-ethylene) nanofibres, which showed a high-performance efficiency for air filtration (99.2% for $PM_{2.5}$ with 64 Pa pressure drop) with a thickness of 5 mm. Qian *et al.* [22] developed a thermal cross-linking method to obtain nanofibre-derived aerogels (NFAs) with 99.9% filtration efficiency for particulate matter$_{2.5}$ ($PM_{2.5}$) and 177 Pa pressure drop. Deuber *et al.* [23,24] found that NFAs could act as deep-bed filters that were capable of handling high dust loadings without any loss in filtration performance or increase in pressure drop. Although these studies provided gelation-free ways for removing $PM_{2.5}$ compared with the traditional filters, the aerogels were often sandwiched between two non-woven mats to avoid airflow damage owing to the weak mechanical properties, limiting the feasibility of actual use.

In the present work, electrospun PI nanofibres are chosen as building blocks due to the excellent thermal-oxidative stability and high mechanical strength [25,26]. By simply dispersing the nanofibres in the mixture of water and tert-butanol and followed by freeze-drying, highly flexible PI nano/microfibre aerogels (PI-N/MFAs) with tunable porous structure and excellent performance for $PM_{2.0}$ filtration were obtained. We focused on reinforcing the skeleton of fibrous aerogels by welding the PI microfibres and nanofibres at their cross-points through solvent-vapour treatment rather than heating and cross-linking, and then fabricated a composite aerogel filter using an *in situ* method. The effect of solvent contents on volume shrinkage rate and compressive resiliency of aerogels were also studied. In such a 3D aerogel, PI microfibres provided the framework support while PI nanofibres were randomly bonded on the microfibres, possessing extraordinary flexibility and elasticity, ultra-low density and high porosity. For a demonstration involving $PM_{2.5}$ removal from air, a composite filter was constructed by immersing a needle-punched non-woven made of polyester (PET) into the suspension of PI, followed by ultrasonic homogenization and freeze-drying. Such a filter showed higher airflow sustainability compared to the filters with surface-deposited nanofibres only.

# 2. Material and methods

## 2.1. Materials and chemicals

Polyimide powder (SX100P) was purchased from Hangzhou SURMOUNT Technology Co., Ltd (Hangzhou, China). Tween 80, *N,N*-dimethylformamide (DMF), *N,N*-dimethylacetamide (DMAc),

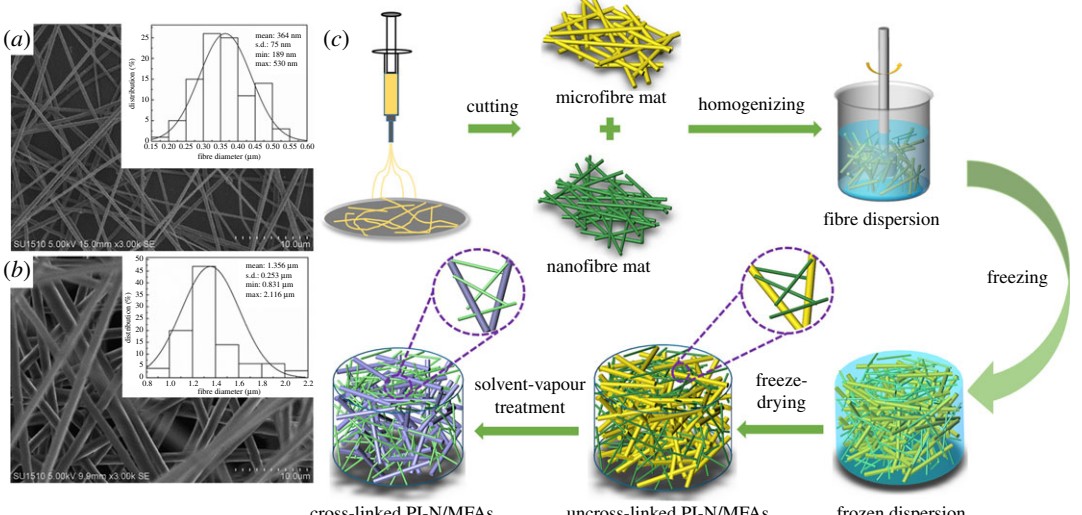

**Figure 1.** SEM images showing the morphologies of PI (*a*) nanofibres and (*b*) microfibres. The insets in (*a*) and (*b*) show the distribution of fibre diameters. (*c*) Schematic illustration showing the fabrication of PI-N/MFAs.

dioxane, tert-butanol and dichloromethane (DCM) were all obtained from Sinopharm Chemical Reagent (Shanghai, China). As noted, all the chemicals were used as received without further purification.

## 2.2. Fabrication of mats of polyimide nanofibres/microfibres

The solution for electrospinning was prepared by dissolving PI powder in a mixture of Tween 80, DMF and DMAc ($v/v/v = 0.25 : 1 : 1$) at a final concentration of 16% ($w/v$). The solution was loaded into a 10 ml plastic syringe with a stainless-steel needle (0.8 mm inner diameter) attached and dispensed by a syringe pump. The injection rate was set to 0.8 ml h$^{-1}$. Nanofibres were spun onto an aluminium foil at a high voltage of 25 kV, with a collecting distance of 20 cm. The temperature and relative humidity during electrospinning were $33 \pm 3°C$ and $55 \pm 5\%$, respectively. Uniform nanofibres with an average diameter of $364 \pm 75$ nm (figure 1*a*) were obtained.

For the PI microfibre mat, the PI solution was prepared by dissolving PI powder in a mixture of dioxane and DMF ($v/v = 1 : 1$) at a final concentration of 25% ($w/v$). The solution was loaded into a 10 ml plastic syringe with a stainless-steel needle (0.8 mm inner diameter) attached and dispensed using a syringe pump. During electrospinning, the injection rate was kept at 1.0 ml h$^{-1}$. Microfibres were collected by an aluminium foil under the conditions of 20 kV for high voltage and 15 cm for collecting distance. The temperature and relative humidity were $33 \pm 3°C$ and $55 \pm 5\%$, respectively. Uniform microfibres with an average diameter of $1.356 \pm 0.253$ μm (figure 1*b*) were obtained.

## 2.3. Preparation of PI-N/MFAs

Figure 1*c* shows the schematic illustration for fabricating the PI-N/MFAs. Briefly, PI nanofibre mats (NFMs) and microfibre mats (MFMs) were separately cut into square pieces of approximately $1 \times 1$ cm$^2$ and dispersed in 100 ml of water/tert-butanol mixture ($v/v = 3 : 1$), followed by further cutting and dispersing in a homogenizer (Fluko FM200) at 13 000 r.p.m. for 25 min to generate a uniform suspension of PI fibres. In this step, PI nanofibres were selected as the major precursors to construct the elastic fibrous networks, and different ratios of PI microfibres (0, 50, 25 and 100 wt%) were introduced into the suspension. The different groups were named as PI-N/MFA$_0$, PI-N/MFA$_{25}$, PI-N/MFA$_{50}$ and PI-N/MFA$_{100}$, respectively. Then, the suspension of PI fibres was freeze-dried to create an uncross-linked PI-N/MFAs under different freezing conditions (−20°C, −80°C and −196°C). The as-obtained PI-N/MFAs were further fumed in the DCM vapour (generated from DCM solutions with different volumes: 1.2, 1.4, 1.8 and 2.0 ml, respectively) for 1 h to obtain the welded PI-N/MFAs.

## 2.4. Preparation of PI nanofibre aerogel composite filters

One piece of needle-punched polyester (PET) non-woven (588.19 g m$^{-3}$, 3.06 mm thickness) was placed in the fibre suspension containing 0.4 wt% PI nanofibres. The mixed suspension was freeze-dried and then treated by DCM vapour to produce the welded PI nanofibre aerogels composite filters (PI-NFACFs).

## 2.5. Characterization

The dispersed PI fibres were cast onto an aluminium foil and dried in a vacuum oven. Then, the fibres and the cross-sections of the aerogels were separately coated with gold at 20 mA for 40 s and then imaged using the scanning electron microscope (SU1510, Hitachi Co. Ltd, Japan). The length of the cut fibres was measured from the scanning electron microscopy (SEM) images using ImageJ software. The chemical compositions of PI-N/MFAs were analysed by Fourier transform infrared (FTIR) spectroscopy (Nicolet IS 10, Thermo Fisher Scientific Co., Ltd, USA) in the range of 400–4000 cm$^{-1}$. X-ray diffraction (XRD) spectra of PI-N/MFAs were recorded using an X-ray diffractometer (D2 PHASER, Bruker AXS GmbH, Germany) with Cu X-ray source and a scanning range of 5–50°. The apparent density ($\rho_a$) of an aerogel was calculated as the weight of the aerogel divided by its volume. The porosity of the aerogel was determined by the following formula:

$$\text{porosity } (\%) - \left(1 - \frac{\rho_a}{\rho_s}\right) \times 100\%, \tag{2.1}$$

where $\rho_a$ and $\rho_s$ are the apparent and skeletal density of the aerogel, respectively. The $\rho_s$ is 1.41 g cm$^{-3}$. The pore size distribution of the aerogel was measured by capillary flow porometry (CFP-1100A, Skei do Will Co., USA) using the bubble-point method.

## 2.6. Mechanical performance

The compression tests were performed on HY-940FS equipped with 100 N load cells. In these tests, cylindrical samples with diameters of approximately 30 mm and lengths of 25 mm were employed, and the strain rate was 3 mm min$^{-1}$. Furthermore, to demonstrate compressive reversibility, 100 loading–unloading fatigue cycles were tested on PI-N/MFAs at a large strain of 80% with a loading rate of 5 mm min$^{-1}$. The tensile test was performed on an Instron KD II-0.05 testing system according to the ISO 1798:2008. Rectangular samples with thicknesses of 10 mm were used, and tensile loading rate was 5 mm min$^{-1}$.

## 2.7. Filtration performance

The filtration efficiency, pressure drop and air permeability of the PI-NFACFs were evaluated using an automated filter tester (LZC-H, Suzhou Hua Da Instrument and Equipment Co., China). The experimental set-up is shown in electronic supplementary material, figure S1. The filtration efficiency (at a flow rate of 84 l min$^{-1}$) and air permeability (pressure drop 100 Pa) were calculated using equations (2.2) and (2.3), respectively.

$$\eta = 1 \frac{C_{\text{down}}}{C_{\text{up}}}, \tag{2.2}$$

where $\eta$ is the filtration efficiency; $C_{\text{down}}$ and $C_{\text{up}}$ are the number concentration of particles at filter downstream and upstream, respectively.

$$R = \frac{Q}{A}, \tag{2.3}$$

where $R$ is the air permeability of the measured materials; $Q$ is the air flow rate under 100 Pa pressure drop; and $A$ is the test area used in this test (50 cm$^2$).

# 3. Results and discussion

## 3.1. Preparation of PI-N/MFA$_0$ with tunable porous structure

Figure 2 shows the morphologies of short PI fibres post dispersion in the water/tert-butanol mixture. The fibres were well dispersed, but with slight tangling. The average fibre length was 81.98 (nanofibre),

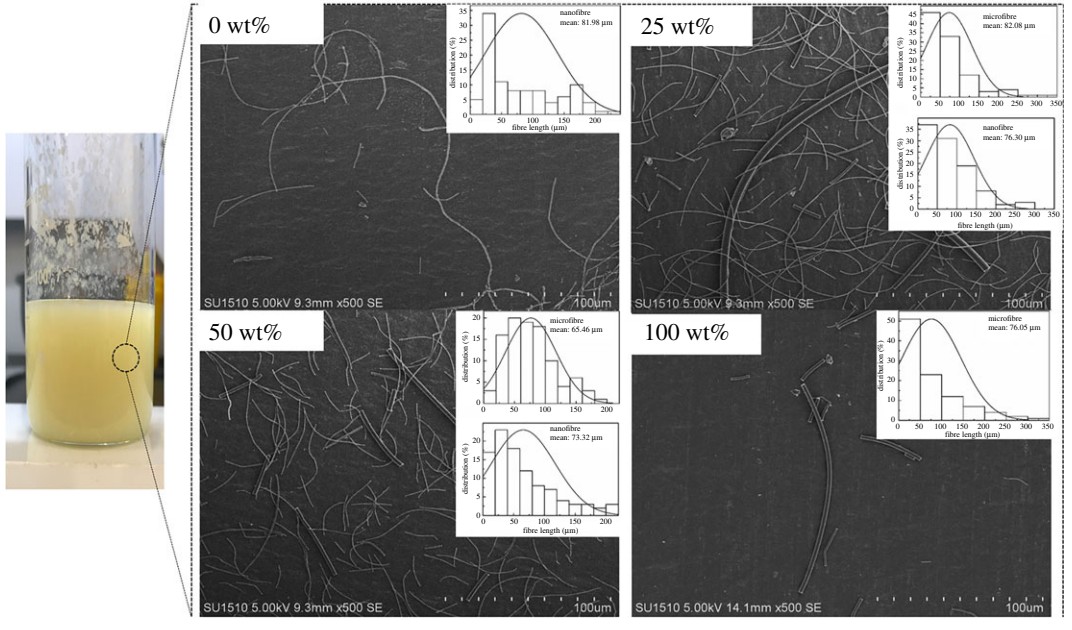

**Figure 2.** Photograph of the dispersion of PI fibres in water/tert-butanol mixture ($v/v = 3 : 1$) and SEM images and fibre length distribution of the dispersed PI fibres with different microfibre contents: 0, 25, 50 and 100 wt%.

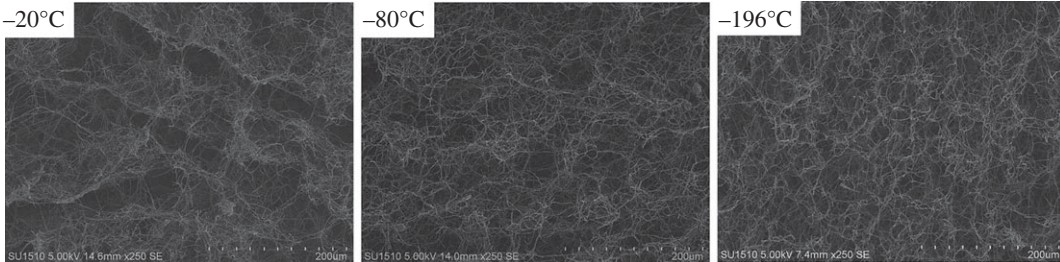

**Figure 3.** SEM images of uncross-linked PI-N/MFA$_0$ frozen under different temperatures: $-20°C$, $-80°C$ and $-196°C$.

82.08/76.30 (nanofibre/microfibre), 65.46/73.32 (nanofibre/microfibre) and 76.05 μm (microfibre), respectively, for the PI-N/MFA$_0$, PI-N/MFA$_{25}$, PI-N/MFA$_{50}$ and PI-N/MFA$_{100}$ samples. Taking PI-N/MFA$_0$ as an example, we can further tune the secondary pores in the as-obtained aerogels by altering the freezing velocity. As shown in figure 3, the secondary pores in variable sizes and shapes were generated due to the formation of water/tert-butanol crystals under different frozen temperatures, which shows consistent results with the previous studies [24].

Owing to the treatment of DCM vapour, the adjacent PI fibres were closely bonded at their cross-points due to the swelling and welding of fibres (figure 4a,b), which contributed to the reinforcement of the aerogel networks. As shown in figure 4c, the PI-N/MFA$_0$ could bear a compressive strain as high as 80% and recover the original volume after the stress was released. The compressive strength and Young's modulus of PI-N/MFA$_0$ were both increased by adding the DCM volumes for generating the vapour (table 1). With the increase of DCM volumes, the entangled nanofibres were welded at a faster pace, and the cellular structure of PI-N/MFA$_0$ changed from porous to dense. In addition, during the compression–release cycles, obvious hysteresis was observed for the aerogels, which could be ascribed to the variation of structures and energy dissipation. After DCM treatment, the PI-N/MFA$_0$ tended to dramatically shrink along the radial direction (figure 4d). The porosity of the aerogel was decreased with the increase of vapour concentration and/or freezing temperature (table 2). Therefore, freezing at $-20°C$ and vapour treating by 1.4 ml DCM were selected for the following test.

According to the FTIR spectroscopy (figure 5a), the characteristic PI absorption bands at 1780 cm$^{-1}$ (C=O asymmetric stretch), 1722 cm$^{-1}$ (C=O symmetric stretch), 1368 cm$^{-1}$ (C-N stretching variation) and 719 cm$^{-1}$ (C=O bending variation in the resulting imide structures) [27,28] all remained after the DCM vapour treatment. The results demonstrate that the treatment of DCM vapour will not change the chemical structures of PI nanofibres. This conclusion was further supported by the XRD analysis (figure 5b).

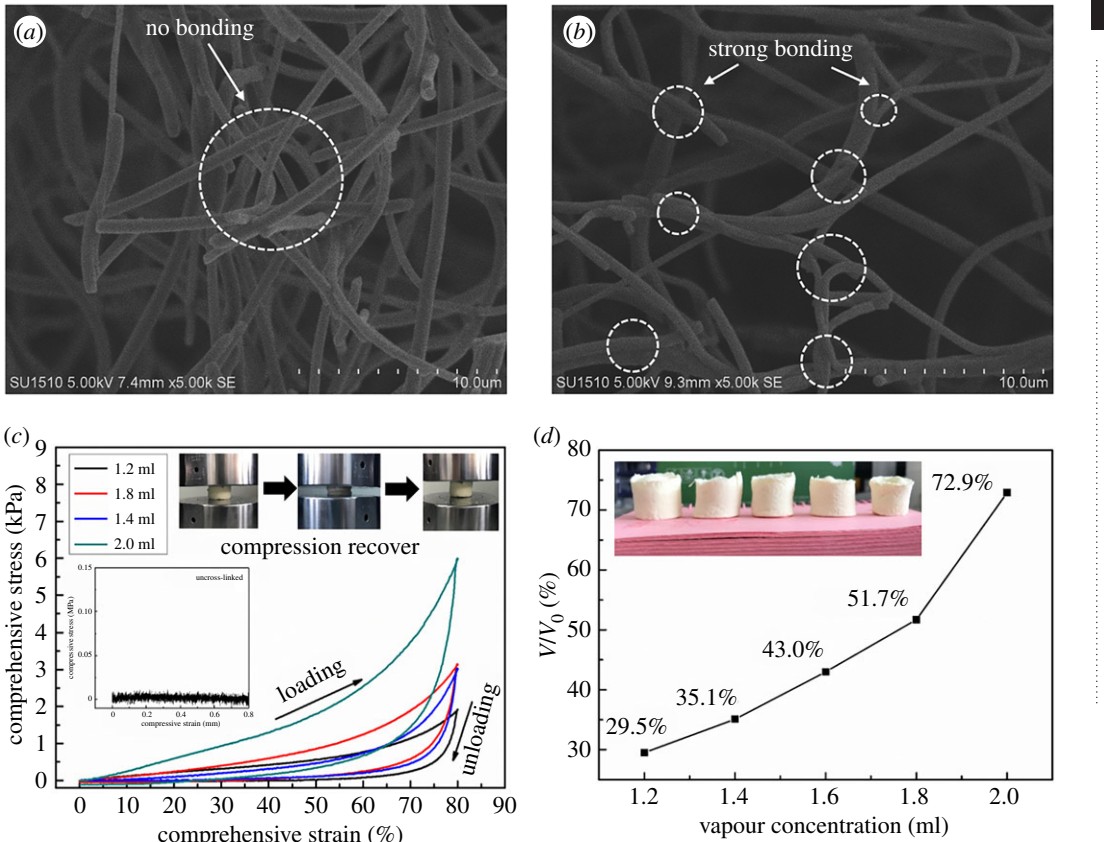

**Figure 4.** SEM images showing the PI-N/MFA$_0$ (*a*) before and (*b*) after treatment by the DCM vapour generated from 1.4 ml DCM. (*c*) Compressive strain-stress curves and (*d*) volume shrinkage behaviours of the PI-N/MFA$_0$ before and after treatment by the DCM vapour coming from different volumes of DCM solutions: 1.2, 1.4, 1.8 and 2.0 ml.

**Table 1.** Mechanical properties of different PI-N/MFA$_0$.[a]

| samples | density (mg cm$^{-3}$) | Young's modulus (kPa) | stress at 80% strain (kPa) |
|---|---|---|---|
| PI-N/MFA$_{0-1.2}$ | 5.25 | $0.60 \pm 0.39$ | 1.95 |
| PI-N/MFA$_{0-1.4}$ | 7.80 | $0.97 \pm 0.026$ | 3.02 |
| PI-N/MFA$_{0-1.8}$ | 9.15 | $1.10 \pm 0.01$ | 3.16 |
| PI-N/MFA$_{0-2.0}$ | 24.03 | $2.44 \pm 0.42$ | 5.99 |

[a]The sample treated by 1.2 ml DCM was named PI-N/MFA$_{0-1.2}$, and so on.

## 3.2. Morphology and structure of PI-N/MFAs

Due to the simplicity of the preparing process and the versatility of electrospun nanofibres, PI-N/MFAs with various shapes could be readily obtained with the use of different moulds (as shown in electronic supplementary material, figure S2). By controlling the amount of PI microfibres (0, 25, 50 and 100 wt%) in the suspension, a density in the range of 4.81–7.12 mg cm$^{-3}$ and a porosity of 99.50–99.66% were produced for the PI-N/MFA$_0$, PI-N/MFA$_{25}$, PI-N/MFA$_{50}$ and PI-N/MFA$_{100}$ samples (table 3). In the networks, PI microfibres acted as a rigid framework to prevent the aerogel from collapsing and shrinking. Significantly, the PI-N/MFAs possessed the highest porosity (99.66%) and lowest density (4.81 mg cm$^{-3}$) when the content of microfibres was 50 wt%.

The SEM images in figure 6 show hierarchically porous architectures in the PI-N/MFAs. Note that the freezing process is crucial to the final architecture of the fibre-derived aerogels. During freezing, the short fibres crowded in the solution were deformed with the growing of solvent crystals, resulting in the generation of fibre-rich and fibre-sparse areas. When the solvent was removed by freezing-drying,

**Table 2.** The porosity of PI-N/MFA$_0$.[a]

| T (°C) | DCM (ml) | | | | | | | | | |
|---|---|---|---|---|---|---|---|---|---|---|
| | 1.2 | | 1.4 | | 1.6 | | 1.8 | | 2.0 | |
| | density (mg cm$^{-3}$) | porosity (%) | density (mg cm$^{-3}$) | porosity (%) | density (mg cm$^{-3}$) | porosity (%) | density (mg cm$^{-3}$) | porosity (%) | density (mg cm$^{-3}$) | porosity (%) |
| −20 | — | — | 5.58 | 99.60 | — | — | — | — | — | — |
| −80 | 5.25 | 99.47 | 7.80 | 99.45 | 8.08 | 99.38 | 9.15 | 99.35 | 24.03 | 98.30 |
| −196 | — | — | 8.60 | 99.14 | — | — | — | — | — | — |

[a]The skeletal density of PI-N/MFA$_0$ is 1.41 g cm$^{-3}$.

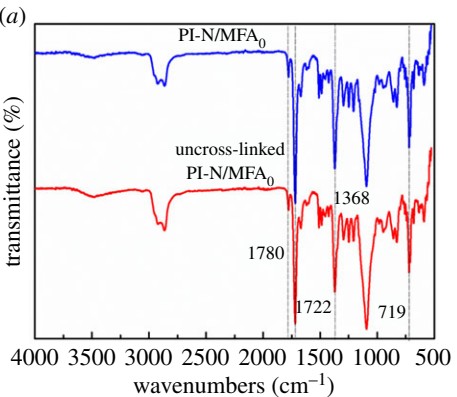
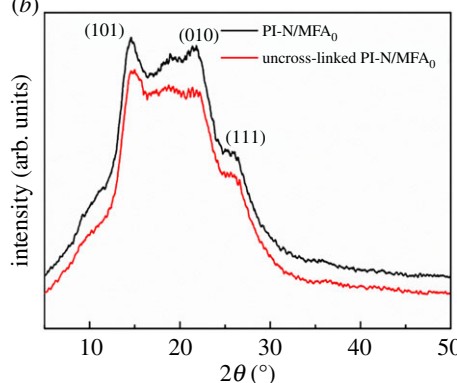

**Figure 5.** (*a*) FTIR spectroscopy and (*b*) XRD patterns of the PI-N/MFA$_0$ before and after treatment by the DCM vapour generated from 1.4 ml DCM.

**Table 3.** The porosity of PI-N/MFAs.[a]

| samples | PI-N/MFA$_0$ | PI-N/MFA$_{25}$ | PI-N/MFA$_{50}$ | PI-N/MFA$_{100}$ |
|---|---|---|---|---|
| shrinkage (%) | 16.58 | 15.64 | 14.90 | 12.82 |
| density (mg cm$^{-3}$) | 5.58 | 5.53 | 4.81 | 7.12 |
| porosity (%) | 99.60 | 99.61 | 99.66 | 99.50 |

[a]The skeletal density of PI-N/MFAs is 1.41 g cm$^{-3}$.

only the fibre skeleton was remained. The fibre-rich areas were fixed as the cell walls while the removal of solvent crystals induced the formation of cellular pores [29–31]. With the increase of the contents of PI microfibres, the cellular structures of PI-N/MFAs were changed from disordered (PI-N/MFA$_0$) to ordered (PI-N/MFA$_{50}$), and finally became uniform (PI-N/MFA$_{100}$) (figure 6$a_1$–$d_1$). The introduction of PI microfibres prevented the networks from shape deformation and position changing in the freezing process. Meanwhile, the growing crystals also pushed the nanofibres towards microfibres, which promoted the entanglement of nanofibres around microfibres and the formation of cell walls with numerous minor pores. The interconnection between major pores and the minor pores thus yielded a truly open-pore network of nano/microfibres with regular cellular characteristics.

## 3.3. Mechanical properties of aerogels

Generally, the particular network structures endow the aerogels with impressive properties (i.e. acoustic, filtration, thermal conductivity etc.), especially the elasticity [32]. Due to the bonding of fibres in the skeletons resulting from both physical entanglement and vapour-induced welding, the PI-N/MFAs exhibited extraordinary flexibility and toughness. A monolith of PI-N/MFAs could recover to its initial dimensions immediately without any fracture when crushed by fingers (electronic supplementary material, figure S3). And the dynamic mechanical performance of the PI-N/MFA$_{50}$ is provided in electronic supplementary material, Movie S1. The hierarchical pores inside the aerogels resulted in excellent mechanical properties under large compressive strains. Figure 7*a* shows the compressive stress–strain curves at a strain maximum of 80% for the samples with various microfibre contents. The stress increased from 2.2 to 6.2 kPa with the microfibre contents increasing from 0 to 50 wt%, while the stress was about 1.6 kPa at 80% compressive when the content of microfibres was 100%. Young's modulus of different aerogels estimated from the elastic regions is shown in figure 7*b*. With the microfibre contents increasing from 0 to 50 wt%, the modulus of the aerogel increased from 0.95 to 3.25 kPa, indicating the stiffness was strengthened. With a further increase in the microfibre content up to 100 wt%, Young's modulus dropped to 0.20 kPa, reflecting a significant decrease in stiffness. Additionally, for aerogels, Young's modulus (and compressive strength) typically scales with density, which was also proved in this study (as shown in table 1). Here, the highest modulus and strength were obtained for PI-N/MFA$_{50}$, which has the lowest density. The height recovery versus the compression ratio of some representative PI-N/MFAs is shown in figure 7*c*. The PI-N/MFA$_{50}$

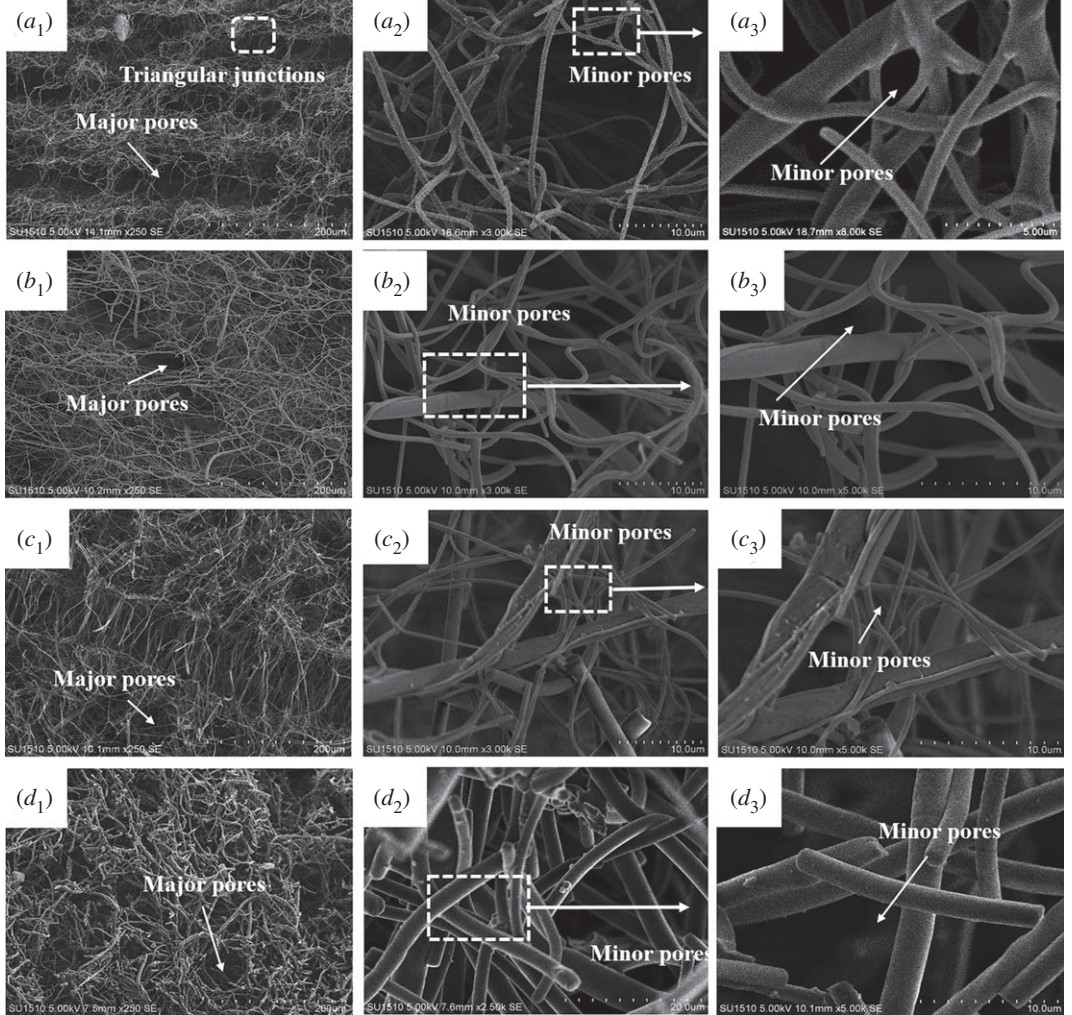

**Figure 6.** The hierarchically porous architectures of PI-N/MFAs with different contents of microfibres. ($a_1$–$a_3$): 0 wt%, ($b_1$–$b_3$): 25 wt%, ($c_1$–$c_3$): 50 wt% and ($d_1$–$d_3$): 100 wt%.

exhibited the highest compression recovery among all the PI-N/MFAs. With an arbitrary compression ratio of 20%, 40%, 60% and 80%, the recovery ratio reached 99.2%, 98.1%, 94.6% and 84.6%, respectively. By contrast, the neat PI-MFA was easy to deform and not able to restore its original appearance in air (figure 7$h$), indicating that the neat PI-MFA exhibited poor resilience. Figure 7$d$ presents the stress–strain curves of the PI-N/MFA$_{50}$ (with a density of 10.4 mg cm$^{-3}$) during the compression–release cycles with a viscoelastic hysteresis, showing that this aerogel was a typically soft and highly deformable material. As shown in figure 7$e$, the PI-N/MFA$_{50}$ exhibits similar or higher recoverable compressive strain and stress relative to the cases involving aerogels made of other types of nanofibres but with a similar density [7,8,22,24,33,34]. In addition, the PI-N/MFA$_{50}$ (with a density of 7.77 mg cm$^{-3}$) also exhibits outstanding cycle performance under a compressive strain of 80% for 100 loading–unloading fatigue cycles as shown in figure 7$f$. It exhibited slight plastic deformation (0.11% at 100th) after 100 cycles, which could be comparable or superior to other counterparts or polymeric foams [6,33].

The tensile $\sigma - \varepsilon$ curves of PI-N/MFAs are presented in figure 7$g$. As the contents of microfibres were increased from 0 to 50 wt%, the breaking load and breaking elongation were firstly increased from 2.174 to 6.206 cN and from 0.269 to 0.702 mm, respectively, but then decreased to 1.602 cN and 0.118 mm when the content of microfibres was 100 wt%. The increased compressive stress, Young's modulus and breaking load could be attributed to the introduction of microfibres, physical entanglements and welding between the nanofibres and microfibres. The high rigidity of PI microfibres resulted in a poor adhesive fastness (as shown in figure 6$d_3$), thus weakening the mechanical strength of the neat PI-MFAs. The overall results confirmed that the PI-N/MFA$_{50}$ exhibits robust mechanical properties compared with the other PI-N/MFAs prepared in this study.

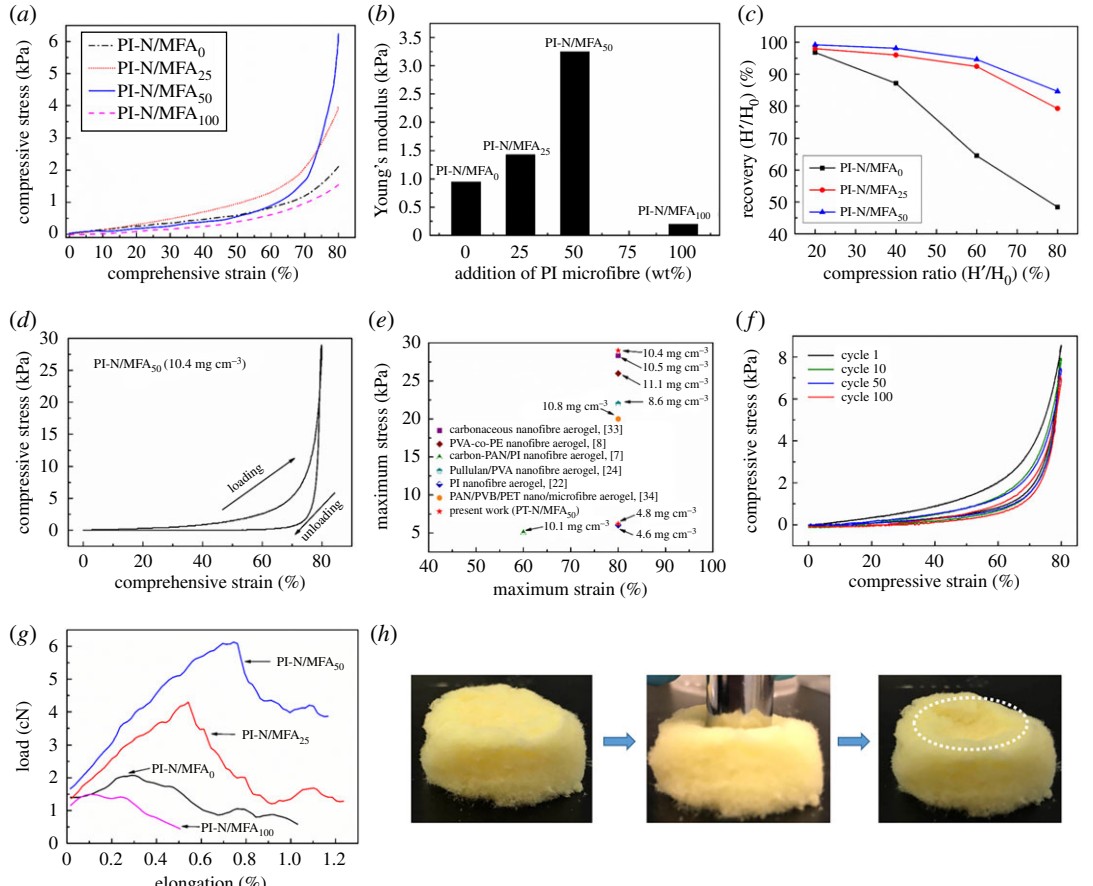

**Figure 7.** Compressive and tensile mechanical properties of PI-N/MFAs. (*a*) Compressive stress–strain maximum of 80% for the aerogel samples with varied amounts of PI microfibres; (*b*) Young's modulus of the PI-N/MFAs; (*c*) compression recovery height versus compression ratio for the PI-N/MFA$_0$, PI-N/MFA$_{25}$ and PI-N/MFA$_{50}$; (*d*) compressive hysteresis of PI-N/MFAs (with a density of 10.4 mg cm$^{-3}$) with 50 wt% PI microfibres at a maximum strain of 80%; (*e*) the recoverable maximum strain and stress of PI-N/MFA$_{50}$ compared with other nanofibre-based aerogels; (*f*) 100 cycles compressive fatigue test with strain of 80%; (*g*) tensile $\sigma - \varepsilon$ curves of PI-N/MFAs with different microfibre contents; (*h*) the images show the compressibility of the aerogel (neat PI-MFA).

## 3.4. Filtration properties of aerogel composite filters

The particle pollution of the air is harmful to the human body. Especially, PM$_{2.5}$ could penetrate into the lung, causing respiratory disease in the people living in the polluted air [35]. Filters with enhanced filtration efficiency is a matter of importance to shut down particulate emission from the source. With the use of hierarchically porous architectures, N/MFAs were regarded as a potential new filter for particle pollution removal. In this study, two methods were adopted to prepare the PI-N/MACFs: ordinary immersion method and ultrasonic impregnation method. Firstly, the morphology and pore size distribution of original PET needle-punched non-woven, PI nanofibre mats composite filter (PI-NFMCF), impregnated PI-NFACF and ultrasonic PI-NFACF were studied (figure 8). It was clearly shown that the fibre length of which PI-N/MACFs exhibited a typically polydisperse distribution ranging from 7 to 127 µm, and the average fibre length was 35.81 µm (figure 8*a*). As shown in figure 8*b*–*e*$_3$, the PI-NFMCF, impregnated and ultrasonic-treated PI-NFACFs showed smaller average pore size and standard deviation relative to the original PET needle-punched non-woven. Especially, the average pore size of PI-NFMCF (1.8 µm) accounted for approximately 5.5% of that of the original PET needle-punched non-woven. The pore size distribution of PI-NFMCF was concentrated, which could be attributed to the formation of a layer of dense nanofibres on the non-woven surface (figure 8*c*). It could be also observed that the pore size of ultrasonic-treated PI-NFACF was smaller than that of the impregnated PI-NFACF, while its pore size was distributed separately. This is mainly because the nanofibres were floated on the surface of PET needle-punched non-woven through impregnating treatment (figure 8*d*), while the short nanofibres of ultrasonic-treated PI-NFACF presented a gradient distribution along the thickness direction (figure 8*e*$_1$). In addition, the nanofibres in the ultrasonic-treated PI-NFACF entangled and formed a 3D network with hierarchical pores (figure 8*e*$_1$–*e*$_3$).

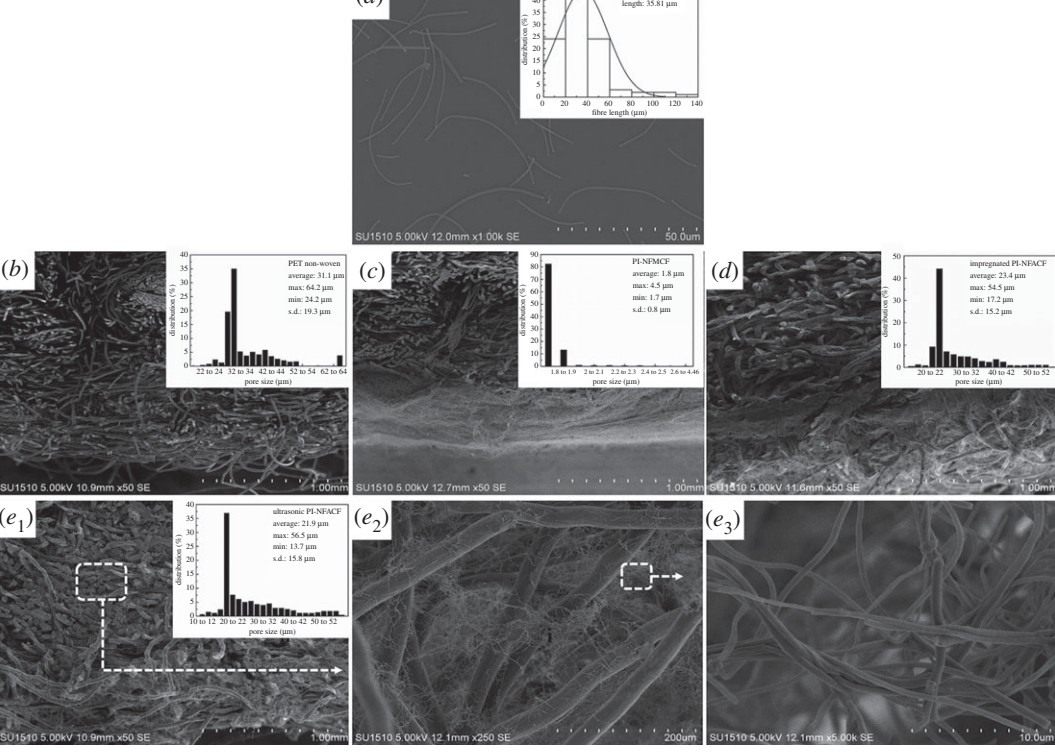

**Figure 8.** The morphology and pore size distribution of PI-NFACFs. SEM images and fibre length distribution of the (*a*) homogenized PI nanofibres, (*b*) original PET needle-punched non-woven, (*c*) PI nanofibre mats composite filter (PI-NFMCF) and (*d*) impregnated PI-NFACF. (*e₁–e₃*) SEM images showing the microscopic architecture of ultrasonic-treated PI-NFACF at various magnifications.

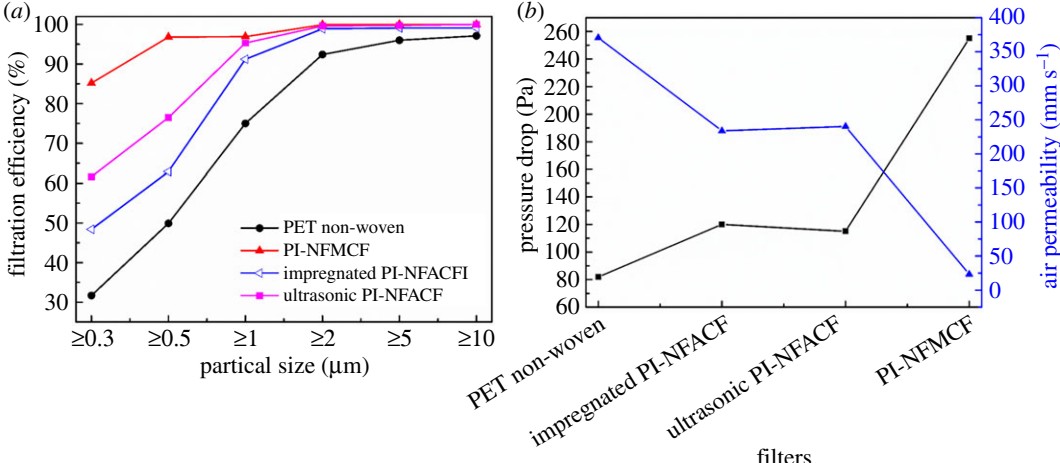

**Figure 9.** The filtration properties of PI-NFACFs. (*a*) The filtration efficiency of PI-NFACFs. (*b*) The pressure drop and air permeability of PI-NFACFs.

Moreover, the filtration properties (filtration efficiency, pressure drop and air permeability) of these composite filters with the same area and mass (133 cm$^2$ and 8.5 g, respectively) were tested (figure 9). Compared with the original PET non-woven, both PI-NFMCF and PI-NFACFs had a modest 15 ∼ 50% increase in filtration efficiency for fine particulate filtration (less than or equal to 1.0 µm, PM$_{1.0}$). We also found that the filtration efficiency of the 0.1 µm particles was significantly higher for the PI-NFMCF and PI-NFACFs than that of the original PET non-woven. Therefore, PM$_{1.0}$ was used to assess the filtration performance. The filtration efficiencies of all filters integrating with nanofibres were more than 90%, while that of PET non-woven was only 75.0%, mainly due to the large difference in the pore sizes of different filters (figure 8). In addition to the composite filters, both PI-NFMCF and ultrasonic PI-NFACF showed ultra-high filtration efficiency (96.9% and 95.3%, respectively) (figure 9*a*). However, the pressure drops of ultrasonic PI-NFACF were 115 Pa, far below

the 255 Pa for the PI-NFMCF (figure 9b). Meanwhile, under equal pressure drop, the filtration efficiency of ultrasonic PI-NFACF was 4.1% higher than that of impregnated PI-NFACF. The excellent filtration performance of ultrasonic PI-NFACF could be ascribed to its hierarchical pore structures (as shown in figure 8$e_1$–$e_3$). A large number of particles were captured on the cell walls while the free air flowed through the secondary pores of the nanofibres in the ultrasonic-treated PI-NFACF, leading to a high filtration efficiency and low-pressure drop. Additionally, the filtration efficiency of ultrasonic PI-NFACF for PM$_{2.0}$ was 99.6%, while the pressure drop was only 115 Pa with a thickness of 3.6 mm.

## 4. Conclusion

In the present work, we have demonstrated a simple method to fabricate ultra-light aerogels from PI nano/microfibres through freeze-drying and subsequent welding induced by solvent-vapour. According to the FTIR and XRD analyses, the solvent-vapour treatment had essentially no significant influence on the chemical structure and crystallinity of the aerogels. The resultant PI-N/MFAs showed low density (4.81 mg cm$^{-3}$), high porosity (99.66%) and tunable cellular structures from disordered to ordered and then uniform. In particular, the secondary pores and mechanical properties of PI-N/MFAs could be varied by tailoring the freezing temperature and the contents of microfibres. The PI-N/MFAs could form ordered cellular structures and improve the mechanical flexibility and toughness with the amount of microfibre increasing to 50 wt%. Meanwhile, by introducing the hierarchical pores into the 3D filtration media, the ultrasonic PI-NFACFs demonstrated a remarkable filtration efficiency for fine particulate filtration (less than or equal to 1.0 μm), and its filtration efficiency for PM$_{2.0}$ reached 99.6% while the pressure drop was only 115 Pa. Furthermore, considering the possibility of composite media consisting of NFAs, even more promising applications could be expected, especially in filtration and separation fields. Besides, the solvent-vapour-induced welding of the nanofibre-based aerogel networks may be extended to other systems to improve the mechanical performances.

Data accessibility. The electronic supplementary figures (experimental set-up for testing filtration efficiency, photographs of PI-N/MFA$_0$ and reversible manual compression of PI-N/MFAs) and movie (the dynamic compressive behaviour of the PI-N/MFA$_{50}$) supporting this article have been uploaded as part of the electronic supplementary material.

Authors' contributions. Y.S., D.L. and B.D. contributed to the conception of the study and design of the experiments, Y.S. performed the experiments, analysis and interpretation of data and wrote the paper, Q.L. and H.L. contributed reagents and analysis of the data, D.L., B.D. and T.W. revised the paper. All authors gave final approval for publication and agree to be held accountable for the work performed therein.

Competing interests. The authors declare no competing interests.

Funding. This work was supported by the Postgraduate Research & Practice Innovation Program of Jiangsu Provence (KYCX18_1823); Fundamental Research Funds for the Central Universities (JUSRP11808); and Fundamental Research Funds for the Central Universities (JUSRP51907A).

Acknowledgements. Instrument Analysis Centre of School of Textiles and Clothing, Jiangnan University, is gratefully acknowledged for all the equipment employed.

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
