## [Reviewer comments · Royal Society Open Science]

Review History

RSOS-190596.R0 (Original submission)

Review form: Reviewer 1

Is the manuscript scientifically sound in its present form?

Yes

Are the interpretations and conclusions justified by the results?

Yes

Is the language acceptable?

No

Is it clear how to access all supporting data?

Not Applicable

Do you have any ethical concerns with this paper?

No

Have you any concerns about statistical analyses in this paper?

Yes

Recommendation?

Accept with minor revision (please list in comments)

Comments to the Author(s)

The author constructed hierarchically porous aerogels welded by solvent-vapor, which is an interesting work and has never been reported. This article falls within the scope of Royal Society Open Science. However, there are some issues need to be addressed.

1. Whether the solvent vapor can permeate into PI aerogel? Please indicate the relative SEM images.
2. The plastic deformation after 100 compressive cycles should be provided rather than only recovery ratio. In addition, the compressive performance need to be improved further to be comparable with the available aerogels obtained by thermo-crosslinking.
3. The filtration efficiency in Fig. 9 should be presented in the form of line chart.
4. The spell and grammar should be improved further

Review form: Reviewer 2

Is the manuscript scientifically sound in its present form?

Yes

Are the interpretations and conclusions justified by the results?

Yes

Is the language acceptable?

Yes

Is it clear how to access all supporting data?

Yes

Do you have any ethical concerns with this paper?

No

Have you any concerns about statistical analyses in this paper?

No

Recommendation?

Accept as is

Comments to the Author(s)

In this paper the authors demonstrate a simple route to polyimide aerogels by freeze drying and subsequent treatment with DCM which allows for the crosslinking of the fibres.

The characterisation and morphology of the structures is thorough demonstrating the change in morphology from single fibres to crosslinked fibres using SEM.

The authors then go on to describe the mechanical properties and finally the filtration properties of the aerogels demonstrating good performance and relatively low drops in pressure.

Overall I believe that this article should be published as is

Decision letter (RSOS-190596.R0)

04-Jun-2019

Dear Miss Shen:

Title: Robust Polyimide Nano/Microfiber Aerogels Welded by Solvent-Vapor for Environmental Applications
Manuscript ID: RSOS-190596

Thank you for submitting the above manuscript to Royal Society Open Science. On behalf of the Editors and the Royal Society of Chemistry, I am pleased to inform you that your manuscript will be accepted for publication in Royal Society Open Science subject to minor revision in accordance with the referee suggestions. Please find the reviewers' comments at the end of this email.

The reviewers and handling editors have recommended publication, but also suggest some minor revisions to your manuscript. Therefore, I invite you to respond to the comments and revise your manuscript.

Because the schedule for publication is very tight, it is a condition of publication that you submit the revised version of your manuscript before 13-Jun-2019. Please note that the revision deadline will expire at 00.00am on this date. If you do not think you will be able to meet this date please let me know immediately.

- 1) A text file of the manuscript (tex, txt, rtf, docx or doc), references, tables (including captions) and figure captions. Do not upload a PDF as your "Main Document".
- 2) A separate electronic file of each figure (EPS or print-quality PDF preferred (either format should be produced directly from original creation package), or original software format)
- 3) Included a 100 word media summary of your paper when requested at submission. Please ensure you have entered correct contact details (email, institution and telephone) in your user account

4) Included the raw data to support the claims made in your paper. You can either include your data as electronic supplementary material or upload to a repository and include the relevant doi within your manuscript

5) All supplementary materials accompanying an accepted article will be treated as in their final form. Note that the Royal Society will neither edit nor typeset supplementary material and it will be hosted as provided. Please ensure that the supplementary material includes the paper details where possible (authors, article title, journal name).

Best wishes,
Dr Laura Smith
Publishing Editor, Journals

On behalf of the Subject Editor Professor Anthony Stace and the Associate Editor Professor Kim Jelfs.

RSC Associate Editor:
Comments to the Author:
(There are no comments.)

RSC Subject Editor:
Comments to the Author:
(There are no comments.)

Reviewer comments to Author:
Reviewer: 1

Comments to the Author(s)
The author constructed hierarchically porous aerogels welded by solvent-vapor, which is an interesting work and has never been reported. This article falls within the scope of Royal Society Open Science. However, there are some issues need to be addressed.

1. Whether the solvent vapor can permeate into PI aerogel? Please indicate the relative SEM images.
2. The plastic deformation after 100 compressive cycles should be provided rather than only recovery ratio. In addition, the compressive performance need to be improved further to be comparable with the available aerogels obtained by thermo-crosslinking.
3. The filtration efficiency in Fig. 9 should be presented in the form of line chart.
4. The spell and grammar should be improved further

Reviewer: 2

Comments to the Author(s)

In this paper the authors demonstrate a simple route to polyimide aerogels by freeze drying and subsequent treatment with DCM which allows for the crosslinking of the fibres.

The characterisation and morphology of the structures is thorough demonstrating the change in morphology from single fibres to crosslinked fibres using SEM.

The authors then go on to describe the mechanical properties and finally the filtration properties of the aerogels demonstrating good performance and relatively low drops in pressure.

Overall I believe that this article should be published as is

Author's Response to Decision Letter for (RSOS-190596.R0)

See Appendix A.

RSOS-190596.R1 (Revision)

Review form: Reviewer 2

Is the manuscript scientifically sound in its present form?

Yes

Are the interpretations and conclusions justified by the results?

Yes

Is the language acceptable?

Yes

Do you have any ethical concerns with this paper?

Yes

Recommendation?

Accept as is

Comments to the Author(s)

The revisions made by the authors are acceptable and I believe this manuscript can be published as is

Decision letter (RSOS-190596.R1)

08-Jul-2019

Dear Miss Shen:

Title: Robust Polyimide Nano/Microfiber Aerogels Welded by Solvent-Vapor for Environmental Applications

Manuscript ID: RSOS-190596.R1

It is a pleasure to accept your manuscript in its current form for publication in Royal Society Open Science. The chemistry content of Royal Society Open Science is published in collaboration with the Royal Society of Chemistry.

On behalf of the Subject Editor Professor Anthony Stace and the Associate Editor Professor Kim Jelfs.

RSC Associate Editor:
Comments to the Author:
(There are no comments.)

RSC Subject Editor:
Comments to the Author:
(There are no comments.)

Reviewer(s)' Comments to Author:
Reviewer: 2

Comments to the Author(s)

The revisions made by the authors are acceptable and I believe this manuscript can be published as is

Appendix A

Dear editor and reviewers:

Thank you for your letter accepting the manuscript entitled “**Robust Polyimide Nano/Microfiber Aerogels Welded by Solvent-Vapor for Environmental Applications**” pending revision. We have revised the manuscript seriously according to the comments and suggestions. We have also attended to the formatting and language of the manuscript according to your suggestions. The revisions and explanations are listed as the follows. The modifications were marked in red in the revised manuscript.

Response to comments by Referee #1

1. Whether the solvent vapor can permeate into PI aerogel? Please indicate the relative SEM images.

√ Thanks for your comments.

It could be verified that the solvent vapor could permeate in the PI aerogels. As shown in Fig 4a and b (in the original manuscript), the SEM images illustrated the internal looking of the aerogels. It could be seen that the adjacent fibers were closely bonded at the cross points after DCM vapor treatment. In addition, supplementation of this conclusion was carried out by relative SEM images which were shown in following Fig. S1.

Fig.S1 SEM images showing the PI aerogel treatment by the DCM vapor: (a) surface SEM image and (b) internal SEM image.

2. The plastic deformation after 100 compressive cycles should be provided rather than only recovery ratio. In addition, the compressive performance need to be improved further to be comparable with the available aerogels obtained by thermo-crosslinking.

√ Thank you for your advice.

Cyclic compression tests with 100 loading-unloading fatigue cycles at a large strain of 80% were further conducted to study the plastic deformation of PI aerogels (Fig.7f, marked in red in our revised manuscript). As shown in the following image (Fig. S2a and b), there is nearly no deformation could be detected after 100 loading-unloading fatigue cycles, corresponding to the compression curve. According to the data of the 100th cycle, the recovery ratio was as high as 99.89%. It was indicated that the PI aerogel possessed great compressive resilient performance.

When the uncross-linked PI aerogel was processed by heating 160 °C for 1h, serious shrinkage occurred on the aerogels (Fig. S2c). The structure and appearance of aerogel were badly destroyed. However, lower temperature could not provide effective bonding between fibers. That is the reason why we choose vapor treatment to weld the aerogels.

Fig.S2 (a) 100 cycles compressive fatigue test with strain of 80%; (b) Photographs of PI aerogel recovered after 100 cycles compressive fatigue test; (c) Photographs of PI aerogels welded by heating at 120 °C/160 °C for 1h.

3. The filtration efficiency in Fig. 9 should be presented in the form of line chart.

√ Thank you for your advice.

Fig.9 was modified into line chart in the revised manuscript.

4. The spell and grammar should be improved further

√ Thanks for your revise.

We have double checked the manuscript and corrected some minor mistakes, marked in red in the article.

Response to comments by Referee #2

In this paper the authors demonstrate a simple route to polyimide aerogels by freeze drying and subsequent treatment with DCM which allows for the crosslinking of the fibers. The characterization and morphology of the structures is thorough demonstrating the change in morphology from single fibers to crosslinked fibers using SEM. The authors then go on to describe the mechanical properties and finally the filtration properties of the aerogels demonstrating good performance and relatively low drops in pressure.

√ Thank you for your comments.

If any other comments to this revised manuscript, please feel free to let me know.

Thank you very much again for your kind help!

Yours sincerely,

Ying Shen